# Evaluating the Growth of Ceria-Modified N-Doped Carbon-Based Materials and Their Performance in the Oxygen Reduction Reaction

**DOI:** 10.3390/nano12173057

**Published:** 2022-09-02

**Authors:** Xin Wen, Ying Chang, Jingchun Jia

**Affiliations:** Inner Mongolia Key Laboratory of Green Catalysis and Inner Mongolia Collaborative Innovation Center for Water Environment Safety, College of Chemistry and Environmental Science, Inner Mongolia Normal University, Hohhot 010022, China

**Keywords:** rare-earth oxide, heteroatom doping, oxygen reduction reaction, Zn–air battery

## Abstract

Owning to their distinctive electronic structure, rare-earth-based catalysts exhibit good performance in the oxygen reduction reaction (ORR) and can replace commercial Pt/C. In this study, CeO_2_-modified N-doped C-based materials were synthesized using salt template and high-temperature calcination methods, and the synthesis conditions were optimized. The successful synthesis of CeO_2_–CN–800 was confirmed through a series of characterization methods and electrochemical tests. The test results show that the material has the peak onset potential of 0.90 V and the half-wave potential of 0.84 V, and has good durability and methanol resistance. The material demonstrates good ORR catalytic performance and can be used in Zn–air batteries. Moreover, it is an excellent catalyst for new energy equipment.

## 1. Introduction

In recent years, increasingly depleting resources and severe environmental pollution have attracted extensive attention worldwide. To better build a community of shared future for mankind, researchers are committed to developing new renewable and clean energy sources to replace traditional energy sources and mitigate environmental pollution and resource shortage [1,2,3,4,5]. Electrical energy is notable for its clean, efficient, economical, and practical characteristics and has emerged as a pivotal research topic [6]. Several means of electrical energy conversion are available. Devices that convert chemical energy into electrical energy are not considerably affected by external factors, such as the weather, season, or climate. Therefore, energy storage devices such as fuel cells [7,8,9,10,11] and metal–air batteries [12,13,14,15,16,17] are typically favored. With reference to electrochemical devices, the cathodic oxygen reduction reaction (ORR) has been extensively investigated because it involves multiple complex steps, slow reaction kinetics, and high overpotential. Therefore, efficient catalysts must be developed to increase the reaction rate [18,19,20]. At present, commercial precious metal Pt-based materials are considered to effectively catalyze the ORR; however, they are limited by high cost and poor stability [21,22,23]. Therefore, we consider enhancing the applicability of the catalysts in terms of reducing the cost and improving catalyst stability to increase their efficiency in the ORR.

Researchers have gradually developed catalysts based on transition metals [24,25,26], carbon materials [27,28], etc., which can replace precious metals. Extensive studies have indicated that C-based materials are typically characterized by a large specific surface area, good conductivity, and low cost. Doping with heteroatoms can result in additional active sites and promote the adsorption of O_2_ on the material surface, thereby improving the ORR rate [29,30]. Metal organic frameworks (MOFs) are innovative inorganic–organic porous materials that contain numerous carbon elements and different heteroelements owing to different inorganic salts. Therefore, they are ideal precursors for preparing heteroatom-doped carbon materials. Catalysts prepared by modulating C-based materials with MOF precursors are characterized by high nitrogen doping levels and rich defects, which provide abundant active sites for the ORR. Therefore, they are significant in catalyzing the ORR [31,32,33,34,35,36,37,38,39].

In recent years, rare-earth metals have been considered for the ORR because of their unique electronic structure and ability to form numerous effective active sites [40,41,42,43,44,45]. Rare-earth metals are primarily oxides with poor conductivity. They are typically combined with C to further improve their conductivity and dispersion. Bai et al. [46] pyrolyzed layered La_2_O_2_CO_3_ and the precursor zeolitic imidazolate framework-67 (ZIF-67) to synthesize the ZIF-67-800@La_2_O_2_CO_3_ catalyzer. According to the electrochemical test results, the catalyst exhibited better ORR catalytic activity and long-term durability than those of commercial Pt/C in alkaline solutions. The enhanced performance is attributed to La_2_O_2_CO_3_, which inhibits the agglomeration of the catalyst, increases the degree of graphitization, and enriches the pore structure. Li et al. [47] evaluated the ORR performance of nine rare-earth-oxide-based catalysts, particularly focusing on Pr_6_O_11_/NC and its applications. Increasing the number of oxygen vacancies promotes the adsorption and transfer of oxygen species, thereby improving the ORR activity. The 4f electronic layer structure of rare-earth elements presents specific properties to the material. Among the rare-earth-group elements, Ce is the first element with 4f electrons, in which exists electron pairs of Ce^3+^ and Ce^4+^. The mutual transformation of Ce^3+^ and Ce^4+^ promotes the activation of reactants and exposes numerous oxygen vacancies to further adsorb oxygen species and effectively catalyze the ORR [42,47,48,49].

Certain inorganic salts exhibit typical crystal grains and structures, and their outer surfaces can be utilized as ideal structural templates. In particular, owing to their favorable thermal stability, inorganic salts, such as NaCl, KCl, and Na_2_SO_4_, can be directly used as templates for precursor carbonization. Following high-temperature calcination, the crystal structures remain unchanged, and the carbonized substances are evenly distributed among the crystals, thereby forming regular target shapes, and exposing additional active sites. Moreover, these inorganic salts are clean and can be easily separated. Therefore, the salt template method has been widely applied for preparing electrochemical catalysts [50,51,52,53,54,55,56].

In this study, zeolitic imidazolate framework-8 (ZIF-8) was used as a carbon and nitrogen source; NaCl was added as a salt template to calcine ZIF-8 and rare-earth metal Ce at high temperatures to ensure that Ce was uniformly loaded onto ZIF-8. The pore structure was modulated by MOFs and NaCl template to improve the specific surface area of the material. The conductivity of the catalyst was further improved by modifying with Ce. Accordingly, an electrocatalytic ORR catalyst CeO_2_–CN with high activity, low cost, and high stability, was designed. The physical properties were characterized using a series of test methods, and the optimal structural composition was selected. The catalyst with the optimal structure was electrochemically tested and applied to Zn–air batteries. The catalyst demonstrated equivalent activity and better stability than those of commercial Pt/C.

## 2. Experimental Section

### 2.1. Chemicals and Reagents

Zn(NO_3_)_2_·6H_2_O, 2-methylimidazole, methanol, Ce(NO_3_)_3_·6H_2_O, and NaCl were all purchased from Sinopharm Chemical Reagent Co., Ltd. (Shanghai, China). All drugs are analytically pure and were not purified.

### 2.2. Synthesis of Materials

A total of 2.5 g Zn(NO_3_)_2_·6H_2_O was mixed with 11 g 2-methylimidazole, dissolved in 500 mL methanol, stirred at room temperature for 24 h to obtain white turbid solution, then centrifuged at 10,000 rpm for 5 min, washed with methanol three times, vacuum dried at 80 °C overnight, and ground for standby [31].

The 0.2 g restrained ZIF-8 was mixed with 0.2 g Ce(NO_3_) _3_·6H_2_O and 4 g NaCl, ground in an agate mortar for 15 min, and then calcined the sample in Ar atmosphere. It was then calcined at 350 °C for 1 h, and then calcined at 800 °C for 3 h. The heating rate was 3 °C per minute. The obtained sample was soaked in ultrapure water overnight, centrifuged and filtered, and vacuum dried at 80 °C overnight to obtain the target catalyst.

### 2.3. Characterization of Physical Properties

Morphologies of the materials were characterized using scanning electron microscopy (SEM, Regulus8100 of Hitachi). The sizes of the materials were further characterized using field-emission transmission electron microscopy (FETEM, Tecnai G2 F20 of FEI). The forms and crystal structures of the synthesized catalysts were analyzed using X-ray diffraction (XRD, RIGAKU Uitima IV). Testing for possible bonding of materials using X-ray photoelectron spectroscopy (XPS, ESCALAB 250Xi, 150 W) was performed. Analysis of specific surface area and pore size distribution of materials using BET (Quantachrome Instruments EVO) results was also performed.

### 2.4. Electrochemical Characterization

The catalyst (5.0 mg) was completely dispersed in a prepared Nafion solution (1 mL) and sonicated for >30 min to form a uniformly dispersed catalyst slurry. The catalyst slurry was then evenly coated on a glassy carbon electrode and dried at room temperature. The ORR electrocatalytic performance of the catalyst was measured using a rotating disk electrode in an oxygen-saturated KOH solution (0.1 M) at a constant temperature. The electrochemical test includes cyclic voltammetry (CV) and linear sweep voltammetry (LSV), etc. The CV test range is −1 V to 0.2 V with two cycles and a potential scan rate of 100 mV/s. The LSV test range is 0.2 V to −1 V with a potential scan rate of 10 mV/s.

A Zn–air battery, which was developed in-house, was analyzed using a battery test system and an electrochemical workstation. A carbon paper–foam nickel composite dripped with catalyst and polished Zn sheet were used as cathode and anode, respectively. A mixture of Zn(Ac)_2_ (0.2 M) and KOH (6 M) was used as the electrolyte.

## 3. Results and Discussion

The precursor ZIF-8 was synthesized using a previously reported method [31]. Zn(NO_3_)_2_·6H_2_O and 2-methylimidazole were mixed and dissolved in methanol. ZIF-8 was obtained via centrifugation, drying, and grinding. The synthesized ZIF-8, Ce(NO_3_)_3_·6H_2_O, and NaCl were ground in an agate mortar to completely mix the reactants and then calcined at a high temperature. NaCl forms a regular crystal structure through calcination, and the remaining reactants grow between the regular crystals formed by the NaCl. After the washing of the NaCl template, N-doped C-based two-dimensional layered materials loaded with CeO_2_ nanoparticles (CeO_2_–CN) were obtained (Figure 1).

Morphologies of the samples formed at different calcination temperatures were investigated using SEM. The images are shown in Figure 1a–c. The samples generated at 600, 800, and 1000 °C maintain the morphology of the Ce-containing material loaded onto the two-dimensional layered C material. The particle size of the Ce material formed at 600 °C is large and that formed at 1000 °C is relatively small. At 800 °C, the particle size of the Ce formed is small with a large loading capacity. It is almost loaded onto the entire lamellar structure and can expose additional active sites. Therefore, calcining at 800 °C optimizes the morphology of the catalyst. The fine morphology of the sample calcined at 800 °C was observed using TEM, and the sample size was measured (Figure 1d). The material exhibits lattice stripe spacings of 1.10, 1.56, and 1.63 nm, corresponding to the (422), (222), and (311) crystal planes of CeO_2_, respectively (Appendix A). Therefore, the material was loaded onto the C carrier in the form of CeO_2_. N and Ce were successfully doped. Moreover, C, N, O, and Ce were uniformly distributed on the material, as confirmed through the elemental mapping (Figure 1e–i) and energy dispersive spectroscopy (Appendix A).

The crystal structure and existing form of the material were analyzed using XRD. In Figure 2a, for the substances synthesized at 600, 800, and 1000 °C, peaks are observed at 2θ = 28.5°, 33.1°, 47.5°, 56.3°, and 76.7° corresponding to the (111), (200), (220), (311), and (331) crystal planes of CeO_2_ (PDF#34-0394), respectively. However, other heteropeaks are observed in the material calcined at 600 °C, possibly because Zn does not evaporate completely at this temperature, and a small proportion of CeO_2_ is agglomerated with Zn, which corresponds to the large particle size of the material observed in the SEM results (Figure 1a). At temperatures ≥ 800 °C, Zn evaporates completely, leaving a substantial amount of CeO_2_ loaded on the carbon carrier. In addition, the average crystallite size of the loaded CeO_2_ was calculated by the XRD peak of the synthesized sample at 800°C using the Scherrer formula, and the results showed that the average crystallite size of CeO_2_ was 5.21 nm (Appendix A), which is almost consistent with the measured result of 5.30 nm in the TEM image (Appendix A).

The composition and surface bonding of the materials calcined at different temperatures were determined using XPS. Considering the entire spectrum (Figure 2b), the substances synthesized at different temperatures reveal the presence of C, N, O, and Ce. It is noteworthy that Zn was present only in the sample calcined at 600 °C and not at other temperatures, which further corroborates with the XRD test results indicating that Zn was not completely evaporated at 600 °C. In the C 1s fine spectrum (Figure 2c), three peaks corresponding to C–C, C–N, and C–O bonds are identified at 284.6, 286.1, and 288.5 eV, respectively [28,47,57]. In the N 1s fine spectrum of the sample synthesized at 800 °C, three peaks corresponding to pyridinic-N, pyrrolic-N, and graphitic-N are observed at 398.2, 400.0, and 403.8 eV, respectively [27,28,47]. However, graphitic-N is not observed for samples obtained at 600 and 1000 °C (Figure 2d), which is possibly because the graphitization degree of the surface is destroyed at extremely low or high temperatures. Graphitic-N considerably promotes the ORR; pyridinic-N and pyrrolic-N belong to defective N structures, which can provide active sites for the ORR [27,28,47,58]. In the O 1s fine spectrum (Figure 2e), three peaks corresponding to metal–O (Ce–O), lattice-O and adsorbed-O are identified at 529.5, 531.4, and 533.3 eV, respectively [49,59,60,61]. In addition, it was confirmed by XRD and XPS results that there is incomplete evaporation of Zn in CeO_2_–CN-600, and some bond energy changes between Zn and Ce in the material due to the presence of strong intermetallic electronic forces, so that the N1s and O1s spectra are shifted. In the Ce 3d fine spectrum (Figure 2f), eight peaks are identified, and their binding energies are located at 882.3, 885.1, 888.7, 898.4, 901.1, 903.4, 907.5, and 916.8 eV. The first five peaks are attributed to Ce 3d_5/2_ and the remaining to Ce 3d_3/2_. The peaks at 885.1 and 903.4 eV correspond to Ce^3+^ and the remaining to Ce^4+^ [42,49,60,61,62,63]. The presence of Ce^3+^ in the crystal can introduce oxygen vacancies for charge compensation. Therefore, the content of trivalent cerium evidently corresponds to that of oxygen vacancies in CeO_2_. [63] An increased oxygen vacancy content enhances the electrocatalytic performance of the corresponding catalyst. The Ce^3+^ content in CeO_2_–CN–800 is significantly higher than that at other temperatures. Therefore, these fine-spectrum results further prove that the material synthesized at 800 °C has better properties than those at other temperatures.

In order to further analyze the changes in specific surface area and pore size distribution of the materials with calcination temperature, BET analysis was performed on the materials. It can be seen from the figure that the N_2_ adsorption–desorption curves of the samples obtained at different temperatures all belong to type IV isotherms with an H4 hysteresis loop, indicating that the calcined materials all have mesoporous structures (Appendix A). The pore size distribution plots showed that the pore sizes of CeO_2_–CN–600, CeO_2_–CN–800, and CeO_2_–CN–1000 were distributed at 3.69 nm, 3.58 nm, and 3.81 nm, respectively. The comparison of the BET specific surface area illustrates that the specific surface area of the material increases with the increase in the calcination temperature, but the difference between the specific surface area of the samples obtained at 800 °C and 1000 °C is not significant (Appendix A).

The ORR performance of the catalysts was evaluated through electrochemical testing. The ORR properties of the samples calcined at different temperatures and those of commercial Pt/C catalysts were tested using CV. No reduction peak is observed in the CV curves of all the samples in an Ar-saturated atmosphere. In contrast, a distinct reduction peak is observed in the CV curves of all samples in an O_2_-saturated atmosphere, indicating that all samples possess ORR catalytic properties (Figure 3a). All synthesized samples and reference Pt/C samples were further tested via LSV in the range of 625–2500 rpm. The ORR performance of CeO_2_–CN–600 is poor, and the limiting current densities of CeO_2_–CN–800, CeO_2_–CN–1000, and Pt/C decrease with an increase in speed during the test (Appendix A). To further compare the ORR performance, the LSV curves of all the samples at 1600 rpm were plotted and analyzed separately. The peak onset potential (E_0_) of CeO_2_–CN–800 approaches 0.90 V, and the half-wave potential (E_1/2_) is approximately 0.84 V, which exceed the corresponding values of the samples obtained at other calcination temperatures and are equivalent to those of commercial Pt/C with an E_0_ of 0.92 V and E_1/2_ of 0.85 V (Figure 3b).

The number of transferred electrons and H_2_O_2_ yields of the ORR were determined using a rotating ring disk electrode (RRDE) device for all samples. Except for CeO_2_–CN–600, whose ORR performance is inadequate, the other catalysts tend to undergo a 4-electron process, and the optimal catalysts CeO_2_–CN–800 and Pt/C exhibit low H_2_O_2_ yields (Figure 3c). In addition, Koutecky–Levich (K–L) curve fitting was performed using the LSV curves of the samples. The K–L curves of all samples at potentials of 0–0.20 V are straight lines (Appendix A). The number of transferred electrons for each catalyst is calculated according to the K–L curve (Figure 3d), which is consistent with the RRDE test results. Therefore, the catalytic process of the prepared catalyst follows a 4-electron path, and the amount of H_2_O_2_ generated is negligible.

To improve the performance of the catalyst for the ORR, other primary synthesis conditions (ratios of Ce salt to precursor and dry matter to NaCl) were modulated. The Ce salt used was Ce(NO_3_)_3_·6H_2_O; the precursor was ZIF-8, and the dry matter mainly included the Ce salt and precursor. Catalysts with Ce salt to precursor ratios of 1:0.5, 1:1, 1:2, and 1:3 were synthesized. In the CV diagram, no oxygen reduction peak is detected in the Ar atmosphere; however, a distinct oxygen reduction peak is observed under O_2_ saturation conditions (Appendix A). The LSV diagram confirms that for a Ce salt to precursor ratio of 1:1, and the ORR exhibits the maximum E_0_ and E_1/2_ (Appendix A). The number of transferred electrons calculated from the K–L curve fitted using the LSV data is 3.9 (Appendix A), and a 4-electron process is observed. Consistent with the RRDE results (Appendix A), the lowest H_2_O_2_ yield is obtained (Appendix A). In the CV diagram of materials with dry matter to NaCl ratios of 1:1, 1:5, 1:10, and 1:50, no oxygen reduction peak appears in the Ar atmosphere; however, a distinct oxygen reduction peak is observed under O_2_-saturated conditions (Appendix A). The LSV diagram further confirms that when the ratio of dry matter to NaCl is 1:10, the ORR catalyst exhibits the maximum E_0_ and E_1/2_ (Appendix A). The number of transferred electrons calculated from the K–L curve is 3.9 (Appendix A). The RRDE test results confirm that the reaction is approximately similar to the 4-electron process (Appendix A), and the lowest H_2_O_2_ yield is obtained in this case (Appendix A). These experiments indicate that among all the analyzed samples, the highest catalytic performance is obtained when the ratio of the Ce salt to precursor is 1:1 and that of dry matter to NaCl is 1:10. Subsequently, the catalysts were synthesized under these optimal conditions.

Catalyst stability is an important indicator for evaluating catalytic performance. Therefore, the durability and methanol tolerance of the optimized catalyst and Pt/C were tested. The durability was determined through an i-t test for 20,000 s in a KOH (0.1 M) solution saturated with O_2_. The current density of CeO_2_–CN–800 decreases by only 0.19 mA·cm^−2^, while that of Pt/C decreases by 0.89 mA·cm^−2^ (Figure 4a). Comparing the LSV diagrams of CeO_2_–CN–800 and Pt/C before and after the 20,000 s test, the peak initiation potential of Pt/C exhibits a significant negative shift, whereas that of CeO_2_–CN–800 is scarcely altered (Appendix A). Furthermore, the methanol tolerance of the catalyst is illustrated by comparing the CV curves and i-t curves before and after adding methanol. The CV curve of Pt/C is significantly altered, whereas that of CeO_2_–CN–800 is almost unchanged (Figure 4b). The i-t test results also showed that CeO_2_–CN–800 showed almost no change in current density before and after the addition of methanol, while Pt/C changed dramatically at the instant of methanol addition after 300 s (Appendix A). These results confirm that the methanol tolerance of CeO_2_–CN–800 is better than that of Pt/C. Therefore, the CeO_2_–CN–800 catalyst exhibits high stability and can function for prolonged durations with minimal impact on the results.

The performance of the CeO_2_–CN–800 catalyst was further evaluated in a Zn–air battery that was developed in-house. Two self-assembled batteries containing the catalyst were connected in series; subsequently, yellow, red, green, and blue light-emitting diode (LED) lamp beads were successfully illuminated (the rated voltage of red and yellow LED lamp beads is 2.0 V and that of the green and blue LED lamp beads is 3.0 V), which proved that the starting voltage of the battery can reach at least 3.0 V (Figure 5a). Electrochemical impedance spectra of the catalysts indicate a low charge transfer resistance (Appendix A). Subsequently, the performance of the battery was evaluated. The discharge polarization curve and power density of the battery indicate that the battery can reach a peak potential of 1.35 V and power density of 65.0 mW·cm^−2^ (Figure 5b). The battery demonstrates good rate discharge performance at different discharge current densities (Figure 5c). The corresponding voltage platforms at 2, 4, 6, 8, and 10 mA·cm^−2^ are 1.28, 1.25, 1.23, 1.20, and 1.18 V, respectively. In addition, constant–current charging and discharging curves (at 2 mA·cm^−2^) were used to estimate the charging and discharging capacity of the battery. The battery potential difference (ΔE) in the first 20 cycles approaches 0.86 V; ΔE tends to be stable after 200 cycles, which proves that the battery exhibits a good charge–discharge cycle ability during operation (Figure 5d).

CeO_2_–CN–800 demonstrates good ORR catalytic performance owing to the following aspects: (1) The introduction of ZIF-8 provides carbon and nitrogen sources and modulates the uniform distribution of the elements on the surface, which subsequently facilitates the formation of nanoclusters with a uniform distribution of metals. (2) When a salt template is added, the material develops a two-dimensional lamellar structure that exposes additional active sites. (3) CeO_2_ contains the conversion of Ce^3+^ and Ce^4+^ electron pairs, which can subsequently form oxygen vacancies that are conducive to the adsorption of oxygen species. Consequently, the catalytic activity for the ORR can be improved. 

## 4. Conclusions

We successfully synthesized a rare-earth oxide CeO_2_-modified N-doped C nanosheet via salt template and high-temperature calcination methods. By optimizing the synthesis conditions, we obtained the highest performing catalyst, namely, CeO_2_–CN–800, which demonstrates good ORR catalytic performance (E_0_ = 0.90 V; E_1/2_ = 0.84 V), excellent stability, and methanol tolerance. Furthermore, it can replace commercial Pt/C. The assembled Zn–air battery exhibits good rate performance and stability when the optimized catalyst is used. This study provides recommendations for synthesizing rare-earth-modified heteroatom-doped C-based catalysts and developing new environmentally friendly energy sources.

## Data Availability

Not applicable.

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
