# Peer review of "Evaluating the Growth of Ceria-Modified N-Doped Carbon-Based Materials and Their Performance in the Oxygen Reduction Reaction"

_nanomaterials, 2022, doi:10.3390/nano12173057_

Round 1

Reviewer 1 Report

This paper deals with the preparation and characterization of ceria-based catalysts for ORR. The topic is interesting and suitable for Nanomaterials readers.

However some aspects should be discussed more:

- In line 146 authors justify the presence of some peaks in XRD results of 600°C sample considering the presence of Zn, not completely evaporated. However, in XPS results there are no traces of Zn, In my opinion a comment is necessary

- Authors highlight in Fig. 2 the presence of graphitic nitrogen, but for the reader is very difficult to see the relative peak, I had to take a leap of faith to see it.

- Line 176-178 In my humble opinion the statement "these fine-spectrum results 176 further prove that the material synthesized at 800 °C has better properties than those at 177 other temperatures"  is not supported by results. A deeper discussion is necessary

- Specific surface area results are mandatory in electrocatalysis results, so  I suggest to the authors  to perform this analysis on their catalysts and to compare them. calcination temperature affects porosity and specific surface area results.

- In line 242-247 authors discuss results about the influence of methanol on electrochemical results. What is the used methanol concentration? Results can dramatically change changing this parameter.

Author Response

Response to comments from the reviewer # 1:

This paper deals with the preparation and characterization of ceria-based catalysts for ORR. The topic is interesting and suitable for Nanomaterials readers.

However, some aspects should be discussed more:

  • In line 146 authors justify the presence of some peaks in XRD results of 600 °C sample considering the presence of Zn, not completely evaporated. However, in XPS results there are no traces of Zn, in my opinion a comment is necessary

Response: We thank the reviewer for her/his comments and suggestions. We have marked the positions of the corresponding peaks of Zn in the entire spectrum and explained them accordingly in the manuscript. (Please see lines 174-176 and Fig. 2b)

  • Authors highlight in Fig. 2 the presence of graphitic nitrogen, but for the reader is very difficult to see the relative peak, I had to take a leap of faith to see it.

Response: We thank the reviewer for her/his comments and suggestions. The graphitic-N at 403.8 V is only present in the sample synthesized at 800 ℃ and its peak is low, we have re-map it to make it look more obvious. (Please see Fig. 2d).

  • Line 176-178 In my humble opinion the statement "these fine-spectrum results 176 further prove that the material synthesized at 800 °C has better properties than those at 177 other temperatures" is not supported by results. A deeper discussion is necessary.

Response: We thank the reviewer for her/his comments and suggestions. We mentioned in the manuscript that graphitic-N considerably promotes the ORR, and only the samples synthesized at 800 °C had graphitic-N. In addition, the content of trivalent cerium evidently corresponds to that of oxygen vacancies in CeO2. An increased oxygen vacancy content enhances the electrocatalytic performance of the corresponding catalyst. The Ce3+ content in CeO2–CN–800 is significantly higher than that at other temperatures. Therefore, we conclude that these fine-spectrum results further prove that the material synthesized at 800 °C has better properties than those at other temperatures.

  • Specific surface area results are mandatory in electrocatalysis results, so I suggest to the authors to perform this analysis on their catalysts and to compare them. calcination temperature affects porosity and specific surface area results.

Response: We thank the reviewer for her/his comments and suggestions. We have added specific surface area and pore size analysis to the manuscript. (Please see lines 200-210, Fig. S10 and Table S3)

  • In line 242-247 authors discuss results about the influence of methanol on electrochemical results. What is the used methanol concentration? Results can dramatically change changing this parameter.

Response: We thank the reviewer for her/his comments and suggestions. We added methanol at a concentration of 1 M. We have stated this in the manuscript. (Please see SI line 32)

Reviewer 2 Report

The article “Evaluating the growth of ceria-modified N-doped Carbon-based materials and their performance in the oxygen reduction reaction” is devoted to the actual problem of obtaining effective materials for hydrogen energy, it is well and detailed written, contains information on the ceria-modified N-doped сarbon-based materials synthesis, a detailed description of the obtained materials structure and their activity in ORR. However, there are a few things to note about the article:

In the Abstract, it is necessary to indicate in more detail about the materials activity

It is necessary to add a study of the obtained materials area by the BET.

In Section 2.3, it is necessary to add information about specific brands of devices on which the study was carried out and their characteristics.

In section 2.4, details of the electrochemical experiment should be added, such as the diameter and material of the working electrode, reference electrode, counter electrode, RRDE characteristics, CV potential ranges, number of cycles, potential sweep rate, etc.,

Based on the microscopy results (Fig. 1), it is necessary to determine the cerium oxide nanoparticles average size

According to the XRD results using the Scherrer formula, it is possible to calculate the average crystallites size of cerium oxide.

Based on the XPS results, it is necessary to determine the nitrogen atomic fraction in the materials and the proportion of each of the nitrogen forms (pyridinic-N and pyrrolic-N, etc.).

Figure 3a shows the range of potentials from -0.2 to 1.2 V- why was this range chose&

It is advisable to transfer some information from Supporting information to the article, for example Experimental details

The figure S9 shows the results of the impedance for CeO2–CN–800, it is necessary to give the equivalent circuit and details of the study (frequency range, etc.)

Author Response

Response to comments from the reviewer # 2:

The article “Evaluating the growth of ceria-modified N-doped Carbon-based materials and their performance in the oxygen reduction reaction” is devoted to the actual problem of obtaining effective materials for hydrogen energy, it is well and detailed written, contains information on the ceria-modified N-doped carbon-based materials synthesis, a detailed description of the obtained materials structure and their activity in ORR. However, there are a few things to note about the article:

  • In the Abstract, it is necessary to indicate in more detail about the materials activity.

Response: We thank the reviewer for her/his comments and suggestions. We have supplemented the abstract with a more detailed description of catalyst activity. (Please see lines 17-18)

  • It is necessary to add a study of the obtained materials area by the BET.

Response: We thank the reviewer for her/his comments and suggestions. We have added specific surface area and pore size analysis to the manuscript. (Please see lines 200-210, Fig. S10 and Table S3)

  • In Section 2.3, it is necessary to add information about specific brands of devices on which the study was carried out and their characteristics.

Response: We thank the reviewer for her/his comments and suggestions. We have added information about specific brands of devices. (Please see lines 108-114)

  • In section 2.4, details of the electrochemical experiment should be added, such as the diameter and material of the working electrode, reference electrode, counter electrode, RRDE characteristics, CV potential ranges, number of cycles, potential sweep rate, etc.,

Response: We thank the reviewer for her/his comments and suggestions. We have added experiment-related details to the manuscript and supporting information. (Please see lines 120-123 and SI lines 28-30)

  • Based on the microscopy results (Fig. 1), it is necessary to determine the cerium oxide nanoparticles average size.

Response: We thank the reviewer for her/his comments and suggestions. We measured an average grain size of about 5.30 nm in the TEM diagram. (Please see Fig. S1b)

  • According to the XRD results using the Scherrer formula, it is possible to calculate the average crystallites size of cerium oxide.

Response: We thank the reviewer for her/his comments and suggestions. We have calculated the average crystallite size of 5.21 nm using Scherrer formula based on XRD results. (Please see Table S2)

  • Based on the XPS results, it is necessary to determine the nitrogen atomic fraction in the materials and the proportion of each of the nitrogen forms (pyridinic-N and pyrrolic-N, etc.).

Response: We thank the reviewer for her/his comments and suggestions. We have added the relevant content to the support information. (Please see Table S4 and Table S5)

  • Figure 3a shows the range of potentials from -0.2 to 1.2 V- why was this range chose&

Response: We thank the reviewer for her/his comments and suggestions. The potential range in the figure is calculated according to the following equation. In the equation, Eθ is the standard electrode potential of the reference electrode, and the standard electrode potential of Hg/HgO at 25 °C is 0.0977 V. In 0.1 M KOH, the pH is 13, and the test electrode (E) range of CV is set to -0.2 V to 1 V. The calculation results show that the potential range is from -0.135 V to 1.065 V, so the potential range is taken as -0.2 V to 1.2 V.

ERHE=E+Eθ+0.059pH

  • It is advisable to transfer some information from Supporting information to the article, for example Experimental details

Response: We thank the reviewer for her/his comments and suggestions. We have adapted some important experimental details from the supporting information to the manuscript. (Please see lines 120-123)

  • The figure S9 shows the results of the impedance for CeO2–CN–800, it is necessary to give the equivalent circuit and details of the study (frequency range, etc.)

Response: We thank the reviewer for her/his comments and suggestions. We have added relevant content to the manuscript. (Please see SI lines 39-40 and Fig. S9b)

Reviewer 3 Report

In this work, the authors prepared the CeO2-modified N-doped C-based materials using the salt template and high-temperature calcination methods, and the synthesis conditions were optimized. The  CeO2–CN–800 was confirmed to demonstrate good ORR catalytic performance and can be used in Zn–air batteries. Some of the following modifications need before publishing this research article:

1. The article's introduction section should be more elaborated to justify the sufficient novelty of this work.

2.      The graphical abstract should be added.

3. In figure 2 (a), XRD peaks should be labeled with respective planes.

4. What is the  main reason behind the shift in the XPS peaks (Fig. 2 (c, d, e, and f )? Please explain on the basis of the ORR performance of each sample.

5. With figure 4 (b), there should be methanol tolerance tests for ORR with  durability tests .

6.What is the main reason and the composition of the elementals behind the good electrocatalytic performance of CeO2–CN-800 with comparison to the  600 and 1000.

7. Similar research on the electrocatalyst can be cited in the appropriate positions. 10.1016/j.carbon.2021.04.028 (Carbon, 179, July 2021, Pages 89-99), 10.1016/j.mtnano.2021.100146 (Materials Today Nano, 17, March 2022, 100146).

8. Fig.S9 caption is need to correct.

9. The as tested electrocatalyst need the post-analysis characterization such as FE-SEM, XRD or XPS.

Author Response

Response to comments from the reviewer # 3:

In this work, the authors prepared the CeO2-modified N-doped C-based materials using the salt template and high-temperature calcination methods, and the synthesis conditions were optimized. The CeO2–CN–800 was confirmed to demonstrate good ORR catalytic performance and can be used in Zn–air batteries. Some of the following modifications need before publishing this research article:

  • The article's introduction section should be more elaborated to justify the sufficient novelty of this work. 

Response: We thank the reviewer for her/his comments and suggestions. In the introduction we mentioned that the present study was formed under the co-shaping of MOF and salt templates with the addition of rare earth elements to enhance the electrical conductivity of the material, which is the highlight of our study and has been marked in the manuscript. (Please see lines 82-85)

  • The graphical abstract should be added. 

Response: We thank the reviewer for her/his comments and suggestions. We have added the graphical abstract to the abstract section. (Please see Abstract)

  • In figure 2 (a), XRD peaks should be labeled with respective planes.

Response: We thank the reviewer for her/his comments and suggestions. We have marked the corresponding planes in the XRD patterns. (Please see Fig. 2a)

  • What is the main reason behind the shift in the XPS peaks (Fig. 2 (c, d, e, and f)? Please explain on the basis of the ORR performance of each sample.

Response: We thank the reviewer for her/his comments and suggestions. We have added the relevant content to the manuscript. (Please see lines 186-190)

  • With figure 4 (b), there should be methanol tolerance tests for ORR with durability tests.

Response: We thank the reviewer for her/his comments and suggestions. We have supplemented the i-t curves of methanol tolerance. (Please see lines 278-281 and Fig. S7b)

  • What is the main reason and the composition of the elementals behind the good electrocatalytic performance of CeO2–CN-800 with comparison to the 600 and 1000.

Response: We thank the reviewer for her/his comments and suggestions. Some physical characterization results confirm that CeO2-CN-800 has good ORR catalytic performance. First, the SEM images showed that the particle size of the Ce material formed at 600 °C is large and that formed at 1000 °C is relatively small. At 800 °C, the particle size of the Ce formed is small with a large loading capacity. It is almost loaded onto the entire lamellar structure and can expose additional active sites. Therefore, calcining at 800 °C optimizes the morphology of the catalyst. Secondly, we mentioned in the manuscript that graphitic-N considerably promotes the ORR, and it can be confirmed by XPS spectra that only the samples synthesized at 800 °C had graphitic-N. In addition, the content of trivalent cerium evidently corresponds to that of oxygen vacancies in CeO2. An increased oxygen vacancy content enhances the electrocatalytic performance of the corresponding catalyst. The Ce3+ content in CeO2–CN–800 is significantly higher than that at other temperatures. Therefore, we conclude that the material synthesized at 800 °C has better properties than those at other temperatures.

  • Similar research on the electrocatalyst can be cited in the appropriate positions. 10.1016/j.carbon.2021.04.028 (Carbon, 179, July 2021, Pages 89-99), 10.1016/j.mtnano.2021.100146 (Materials Today Nano, 17, March 2022, 100146).

Response: We thank the reviewer for her/his comments and suggestions. We have gained a lot from these two papers and have added them to the manuscript as references. (Please see lines 372-373)

  • S9 caption is need to correct.

Response: We thank the reviewer for her/his comments and suggestions. We have corrected the error (Please see SI line 100)

  • The as tested electrocatalyst need the post-analysis characterization such as FE-SEM, XRD or XPS.

Response: We thank the reviewer for her/his comments and suggestions. We have added analysis to the relevant test results. (Please see lines 162-166, 174-176, 186-190, 193-198 and Fig. S1)

Round 2

Reviewer 1 Report

Accept

Reviewer 2 Report

The authors responded to the comments in full and the article can be accepted for publication.

Reviewer 3 Report

All the comments are answered properly.